# Biomarkers of a Healthy Nordic Diet—From Dietary Exposure Biomarkers to Microbiota Signatures in the Metabolome

**DOI:** 10.3390/nu12010027

**Published:** 2019-12-20

**Authors:** Rikard Landberg, Kati Hanhineva

**Affiliations:** 1Division of Food and Nutrition Science, Department of Biology and Biological Engineering, Chalmers University of Technology, 412 96 Gothenburg, Sweden; kati@chalmers.se; 2Institute of Public Health and Clinical Nutrition, University of Eastern Finland, P.O. Box 1627, 70210 Kuopio, Finland

**Keywords:** Nordic diet, biomarkers, metabolomics, personalized nutrition, microbiota

## Abstract

Whole diets and dietary patterns are increasingly highlighted in modern nutrition and health research instead of single food items or nutrients alone. The Healthy Nordic Diet is a dietary pattern typically associated with beneficial health outcomes in observational studies, but results from randomized controlled trials are mixed. Dietary assessment is one of the greatest challenges in observational studies and compliance is a major challenge in dietary interventions. During the last decade, research has shown the great importance of the gut microbiota in health and disease. Studies have have both shown that the Nordic diet affects the gut microbiota and that the gut microbiota predicts the effects of such a diet. Rapid technique developments in the area of high-throughput mass spectrometry have enabled the large-scale use of metabolomics both as an objective measurement of dietary intake as well as in providing the final readout of the endogenous metabolic processes and the impact of the gut microbiota. In this review, we give an update on the current status on biomarkers that reflect a Healthy Nordic Diet or individual components thereof (food intake biomarkers), biomarkers that show the effects of a Healthy Nordic Diet and biomarkers reflecting the role of a Healthy Nordic Diet on the gut microbiota as well as how the gut microbiota or derived molecules may be used to predict the effects of a Healthy Nordic Diet on different outcomes.

## 1. Introduction

Diet is one of the most important modifiable lifestyle factors contributing to the prevention of non-communicable diseases such as type 2 diabetes, cardiovascular disease and cancer and their main risk factors [1]. Although challenging, alteration towards a healthier diet can delay or even prevent the onset of disease and prevention is considered to be the most sustainable and cost-effective way to manage chronic diseases [2,3]. Traditionally, research has mainly focused on the role of nutrients and specific food items on human health but more recently, dietary patterns that account for interactions between different foods and food components have been widely studied and found strongly linked with health [3,4]. Many studies have suggested that better adherence to healthy dietary patterns, such as the Mediterranean Diet Score (MDS), Dietary Approaches to Stop Hypertension (DASH) score, and Healthy Eating Index (HEI), significantly reduces the risk of cardiovascular disease (CVD) and type 2 diabetes (T2D) [5,6,7]. Further, large randomized controlled studies have shown beneficial effects on hard endpoints and major risk factors of these conditions [8,9,10], although there are large differences between individual responses to such diets [11,12].

Among dietary patterns, a Healthy Nordic Diet has received much attention as a promising habitual diet with comparable health benefits to the Mediterranean diet, first demonstrated by Olsen et al. in 2011 in a landmark paper [13]. In this study, the authors found that a high vs. low adherence to the Healthy Nordic Food Index (HNFI) was associated with an 11% lower mortality in a Danish population. Subsequent observational studies have found that higher reported adherence to Healthy Nordic Dietary indices, e.g., the HNFI [13] and the Baltic Sea Diet Score (BSDS) [14], were inversely associated with, e.g., total mortality [13,15] and abdominal obesity [16]. As any dietary pattern, the Healthy Nordic Diet differs somewhat across regions but it is a dietary pattern that complies with current dietary guidelines and includes traditional Nordic food items, such as vegetables, fish, fruits, whole grains (oats and rye), and various seafoods [17,18,19]. In randomized control trials [10,20,21,22], the Healthy Nordic Diet has been shown to have similar beneficial effects on cardio-metabolic risk factors to the Mediterranean diet [21] but other studies have shown only minor effects in some outcomes measured [10,22,23].

There are different potential explanations to the mixed results from the intervention studies and from some of the observational studies. Dietary assessment could be one part of the explanation, since it is one of the greatest challenges in nutritional studies [24,25]. In observational studies, slightly different food items have been included and different scoring systems have been used to capture the Healthy Nordic Diet [13,26,27]. Moreover, it is challenging to capture the specific food intakes reliably over the relevant time period and the questionnaires have not been optimized to capture healthy Nordic food items. Moreover, intervention studies with the Nordic diet have typically provided slightly different foods and offered some degrees of freedom in the choices of exact food items, which further contribute to the difficulties in exact intake assessment [10,21,22,23,28]. The use of specific dietary biomarkers that reflect the Healthy Nordic Diet pattern overall, as well as individual food items included in this diet, could be useful objective instruments to improve the analysis. Such markers can also be used to address compliance in dietary intervention studies.

Another aspect that may explain different results across studies, and particularly for intervention studies may be the fact that individuals respond differently to the same diet. Results from intervention studies of a Nordic Healthy Diet have shown that there is a large heterogeneity in response to the dietary intervention within and between studies. All studies so far have been conducted under the general “one-size-fits-all” population-based paradigm, where all subjects receive the same intervention irrespective of their health status, age, life-style, genetic background and gut microbiota. However, recent research has shown that those intrinsic and external factors contribute as determinants of the response/non-response to a Healthy Nordic Diet, or individual foods included in such diet, on several outcomes such as blood glucose levels [29,30], body weight [31,32] and blood lipids [33,34].

The aim of this narrative review is to highlight the most recent findings and provide the current state of the art of biomarkers that reflect a Healthy Nordic Diet and to provide an overview of the recent developments in personalized nutrition with focus on the search for biomarkers that define responders and non-responders to a Healthy Nordic Diet or food components thereof. We also suggest future steps to be taken to develop our understanding of the health effects of the Healthy Nordic Diet and its use in personalized nutrition.

## 2. Classification of Biomarkers

There are several different types of biomarkers reflecting biological effects, susceptibility and exposures. Dietary biomarkers is a group of exposure biomarkers that reflect the intake or efficacy of a specific food item, nutrient or dietary pattern, depending on whether the biomarker is a compound resulting from the consumed dietary item, or if it is an endogenous metabolite reflecting the change in the host metabolic homeostasis evoked by the diet. A comprehensive classification system [35] and a validation scheme for Biomarkers of Food Intake (BFIs) have recently been developed by the European FoodBall consortium [36]. According this system, biomarkers can be divided into six classes: food compound intake biomarkers (FCIBs), biomarkers of food or food component intake (BFIs), dietary pattern biomarkers (DPBs), food compound status biomarkers (FCSBs), effect biomarkers, physiological or health state biomarkers. In traditional classification of dietary biomarkers, FIBs can be classified as recovery and concentration biomarkers depending on their characteristic [37]. Recovery biomarkers reflect the amount of a compound excreted compared with its intake on an absolute scale over a specific time period, whereas concentration biomarkers reflect the correlation between the concentration of a compound in a biological sample with the corresponding intake [38]. Recovery biomarkers are regarded as a gold standard and can be used for the calibration of other dietary assessment methods [39]. Unfortunately, only very few recovery biomarkers are currently available. Sometimes prediction biomarkers are mentioned as a third category, falling in between the recovery biomarkers and concentrations biomarkers [40]. Most biomarkers belong to concentration biomarkers [37]. These biomarkers typically represent a diet-specific molecule or a metabolite thereof, measured in blood, urine, adipose tissue or feces. Concentration biomarkers increase in response to the specific food intake but are also affected by non-dietary determinants such as BMI, age, gender, etc., to various extents. Therefore, concentration biomarkers cannot be translated to reflect absolute intake but only a correlation with true intake.

## 3. Discovery of Biomarkers of a Healthy Nordic Diet Using Metabolomics

Much effort has recently been expended to discover new biomarkers of specific food items including foods in a Healthy Nordic Diet, using emerging metabolomics techniques. Metabolomics is a valuable tool for identifying metabolites that could objectively reflect specific food exposures [41,42,43] or dietary patterns [44] within the context of a Healthy Nordic Diet. Metabolite biomarkers could provide a complement to self-reported dietary assessments and thereby aid the understanding of disease-related metabolic processes influenced by diet [35,36,37]. Biomarkers may be discovered in different sample matrices such as plasma, urine, adipose tissue, hair and nail clippings using targeted or untargeted approaches [37,38,39,40,45,46,47]. In targeted metabolomics, a defined set of well-characterized and annotated metabolites are analyzed typically in quantitative platforms such as triple quadrupole mass spectrometry (QQQ-MS), utilizing pure chemicals as standards. Targeted metabolite analyses have been used to analyze compounds known or suspected to be putative biomarkers of specific foods included in the Healthy Nordic Diet such as whole grain wheat, rye and oats, legumes, apples and pears, berries, fatty fish and dairy. For example, odd-numbered alkylresorcinols have been discovered, validated and used as biomarkers of whole grain wheat and rye [48], even-numbered alkylresorcinols have been used to reflect quinoa intake [48] and avenanathramides and avenacosides have been suggested as biomarkers of oat intake [48]. Individual biomarkers that reflect specific cereal foods have been detected in plasma, urine and adipose tissues and by utilizing chemometric, multivariate tools, there are new possibilities to use combinations of several biomarkers, i.e., biomarker panels, which may improve the prediction of outcomes as well as the monitoring of compliance or measuring food intake compared with single concentration biomarkers [48].

In contrast, untargeted approaches aim at maximizing the metabolite coverage in sets of biological samples, even though the vast majority of measured metabolic features remain unidentified. A common analytical platform for profiling assays is quadrupole time-of-flight mass spectrometry (QTOF-MS) with chromatographic separation in either liquid or gas phase, or Nuclear Magnetic Resonance (NMR)-based platforms. Inherent to the wide coverage, untargeted approaches are well suited for exploratory biomarker studies, and this approach has been used to mine for dietary exposure biomarkers reflecting, e.g., total or specific whole grain intake as well as specific grain-based foods after controlled interventions with specific foods or reported food intakes [48]. Untargeted and targeted approaches are complementary to each other and are both required to discover and validate dietary biomarkers, respectively. A typical workflow involves the identification of putative biomarker candidates via an untargeted profiling approach followed by validation of the biomarkers in targeted, quantitative analyses applied preferentially in other study cohorts. Recently, Zhu et al. [49] combined untargeted and targeted metabolomics approaches to discover biomarkers of whole grain wheat intake in urine samples after intake of whole grain wheat bread vs. refined wheat bread in a kinetic study in 12 subjects. A panel of urinary markers consisting of seven alkylresorcinol metabolites and five benzoxazinoid derivatives as specific biomarkers along with five phenolic acid derivatives were suggested to reflect whole grain wheat intake. Panels of biomarkers of whole grain, refined grain or fractions of specific grains appear promising but remain to be evaluated in larger studies. Another interesting group of compounds associated with whole grain intake, with potential implications related to endogenous energy metabolism, are the betainized compounds recently reported [45] and validated with quantitation [46].

## 4. Biomarkers of Food Intake and Metabolic Effects Reflecting a Healthy Nordic Diet

To date, only few valid food intake biomarkers exist for specific food items and only a handful of studies that have investigated and suggested biomarkers that reflect a specific dietary patterns [42,43,46]. Based on different dietary intervention studies with a Healthy Nordic Diet and from one observational study where adherence to a Healthy Nordic Diet was assessed, biomarkers reflecting intake have been suggested (Table 1). Many of these biomarkers are of limited use when used in separate since they are not specific.

In a study by Andersen et al. [47], the investigators aimed to find biomarkers of compliance to a New Nordic Diet (NND) vs. Average Danish Diet (ADD), and applied LC-QTOF-MS based metabolomics on 24 h urine samples collected during and after a 6 month parallel dietary intervention study [50], with these two diets provided to 181 over-weighted men and women with one additional feature of the metabolic syndrome. The authors found 52 metabolites that explained clustering of reported food intakes. Based on the urinary metabolome, it seemed that the ADD was better reflected in urine than the NND. A higher number of metabolites from ADD were found, and they reflected both specific to individual foods, which are only allowed in the Danish diet, and represent more general features of the diet such as higher intakes of animal protein and heat-treated foods. For the NND, the metabolites found were mainly reflecting a high intake of fish, fruit, and vegetables. The authors speculated that the reason for why fewer metabolites were found for the NND was because it is a seasonal diet. Overall, it appears an open question to what extent urinary metabolomics could be used to address compliance to NND, since rather few metabolites were direct related to high adherence to such a diet and most of them are not specific to the foods consumed as part of the NND. Further studies are needed to test the performance in an independent population.

In another study, Kakhimov et al. [41] analyzed fasting plasma samples from a subset of 141 individuals from the study by Andersen et al. [47]. The aim was to investigate the long-term metabolic effects of the NND and to assess the effects of weight loss (obtained through the study design), gender and season on the Gas chromatography-mass spectrometry (GC–MS) based metabolome. Significant and novel metabolic effects of the diet, resulting weight loss, gender, and intervention study season were revealed using two multivariate approaches, partial least square-discriminant analysis (PLS-DA) and Analysis of variance-simultaneous component analysis (ASCA). Several metabolites reflected differences in the diets, particularly plant foods and seafood. Moreover, ketone bodies related to energy metabolism and gluconeogenesis differed between the intervention groups. In total, 33 metabolites discriminated NND from the ADD—of which, 21 were identified (Table 1). Metabolites found to be present at higher concentrations in subjects on the NND could be categorized into two groups—those directly reflecting NND with a higher intake of fish, vegetables including crucifers, as well as whole grain; or metabolites that mainly reflected the impact of NND on energy metabolism to increase gluconeogenesis and ketosis, and subsequent improvement in insulin sensitivity. These metabolites are altogether interpreted as metabolites with health-beneficial effects. The study strongly indicates that healthy diets high in fish, vegetables, fruit, and whole grain help to improve insulin sensitivity by increasing ketosis and gluconeogenesis in the fasting state and that some of the foods and metabolic processes may further explain the reduced blood pressure and cholesterol, thereby generally improving metabolic health. The study strongly indicates that healthy diets high in fish, vegetables, fruit, and whole grain facilitated weight loss and improved insulin sensitivity by increasing ketosis and gluconeogenesis in the fasting state.

In a third metabolomics study based on plasma samples from the original weight loss study with a NND [50], Acar et al. [42] investigated the effects of the NND vs. ADD as described above on the plasma metabolome using LC-QTOF-MS. They found that supervised machine learning with feature selection can separate NND and ADD samples with a good performance (AROC (area under the receiver-operating curve) = 0.88). The NND plasma metabolome was characterized by diet-related metabolites, such as pipecolic acid betaine (whole grain), trimethylamine oxide, and prolyl hydroxyproline (both fish intake), while theobromine (chocolate) and proline betaine (citrus) were associated with the ADD. Regarding effect markers, the authors also found differences in amino acid and fat metabolism that characterize AAD whereas higher concentrations of polyunsaturated phosphatidylcholines were characteristic for the NND. The authors concluded that the plasma metabolite profiles were predictive of dietary patterns and reflected good compliance while indicating the effects of potential health benefit, including changes in fat metabolism and glucose utilization.

In a study by Hanhineva et al. [43], LC-QTOF-MS metabolomics was applied to fasting plasma samples of 106 men and women with metabolic syndrome after the consumption of whole grains, whole grains + fatty fish + bilberries or a refined wheat diet in order to discover biomarkers reflecting the specific diets or food items in the diets and effect biomarkers. Among the 3130 molecular features collected in the four different Liquid Chromatography-Mass Spectrometry (LC-MS) modes (i.e., the chromatography was conducted in the reversed phase and normal phase (hydrophilic interaction) mode and both negative and positive ionization were applied), the authors found that a total of 400 were significantly changed after the intervention with either one or both intervention diets when compared with the fold-change values obtained from the control group (Student’s *t*-test, *p* < 0.05). The corresponding figure after correction for multiple testing was 90 metabolite features. Two robust biomarkers after whole grain intake were identified as glucorunidated alkylresorcinols (specific for whole grain wheat and rye) and a betainized compound. For the whole grain+ fatty fish + bilberry group, several biomarkers were found including lipid compounds and 3-carboxy-4-methyl-5-propyl-2-furanpropanoic acid (CMPF). CMPF was found to be related to fatty fish intake, but not with other food items and alkylresorcinol metabolites were highly correlated with whole grain wheat and rye intake, thus reflecting these aspects of the diets. Moreover, alterations in the endogenous metabolism of betaine, amino acids, and lipids was observed and indicate the potential beneficial metabolic effects of a Healthy Nordic Diet rich in whole grains, fatty fish, and berries.

In a recent study by Tuomainen et al. [46] a new Liquid Chromatography- triple Quadrupole Mass Spectrometry (LC-QQQ-MS) method targeted for betainized compounds was developed and applied on fasting plasma samples from the SYSDIET intervention study to evaluate the effects of the intervention as well as correlate the concentrations of betainized compounds with measures of metabolic health. SYSDIET is a 18–24 week multi-center study with the aim to evaluate the metabolic effects of a Healthy Nordic Diet vs. a control diet reflecting the average Nordic diet among subjects (166 completed the study) with metabolic syndrome [10]. Among the compounds measured, pipecolic acid betaine was the only one that differed significantly after the intervention (higher in the Healthy Nordic Diet group). This compound was inversely associated with fasting plasma insulin, IL-1 receptor antagonist and serum LDL/HDL cholesterol. The authors concluded that further studies are needed to elucidate the detailed biological function of pipecolic acid betaine and other betainized compounds.

So far, only one observational study has reported on biomarkers reflecting a Healthy Nordic Diet [44]. In this study, Shi et al. used untargeted metabolomics to identify metabolites related to a priori-defined Healthy Nordic Dietary indices, Baltic Sea Diet Score (BSDS), and Healthy Nordic Food Index (HNFI), and evaluated their associations with the T2D risk in a case-control study nested in a Swedish population-based prospective cohort. In total, plasma samples from 421 case-control pairs at baseline and samples from a subset of 151 healthy controls at a 10 year follow-up were analyzed using an LC–MS-based metabolomics workflow [51]. Index-related metabolites were identified using random forest followed by partial correlation analysis adjustment for lifestyle confounders. Metabolite patterns were derived using principal component analysis (PCA). Odds ratios (ORs) of T2D were estimated using conditional logistic regression. Associations were also assessed for 10 metabolites previously identified as linking a Healthy Nordic Diet with T2D. In total, 31 metabolites were associated with BSDS and/or HNFI (−0.19 ≤ *r* ≤ −0.21, 0.10 ≤ Intra-Class Correlation Coefficient ≤ 0.59). Among index-related metabolites, five were associated with both indices: docosahexaenoic acid (DHA), lysophatidylethanolamine (lysoPE 22:6), γ-tocopherol, and two unknown metabolites. Metabolites that were positively associated with the indices were also positively correlated with whole grains, fruits, vegetables, and fish, and negatively correlated with red/processed meat and total fat. The opposite was observed for metabolites that were inversely associated with the indices. Two PCs were determined from index-related metabolites: PC1 reflective of adherence to the indices (*r* = 0.27 and 0.25 for BSDS and HNFI, ICC = 0.45) was not associated with T2D risk. PC2 was weakly related with indices but had a stronger correlation with foods not part of indices, e.g., pizza, sausages and hamburgers. PC2 was also significantly associated with T2D risk. Predefined metabolites were confirmed to be reflective of the consumption of whole grains, fish, or vegetables, but were not related to T2D risk.

## 5. The Gut Microbiota as a Predictor of Responsiveness to a Healthy Nordic Diet

The gut microbiota is an essential part of the human digestive system and has a large impact on diet-derived nutrients and other compounds entering human body [52]. The human gut is the habitat for thousands of bacterial species representing various phyla such as Bacteroidetes (Gram-negative), Firmicutes (Gram-positive), and Actinobacteria (Gram-positive), with hundreds of different genera and individual species [53,54]. The past decade of research has proven that this complex microbial community inhabiting our digestive tract, mainly colon, has numerous roles, such as absorption and metabolism of compounds, and development of immune response, and this large impact on host physiology and metabolism makes it important influencer in terms of human health. Scientific evidence within the area is accumulating, and so far, gut microbiota composition has been related to the risk of obesity [54], type 2 diabetes [55], cardiovascular diseases [56,57], immunological diseases [58], inflammatory bowel disease [59], and gastrointestinal cancer [60].

It has recently been shown that the environment affects the gut microbiota to a greater extent than genetic properties [61] and the gut microbiota is a target for lifestyle interventions, including diet, aiming at modulating its composition and metabolic outcome. Gut microbiota and diet interactions could affect susceptibility, as well as offering a means for the prevention and treatment of diseases [52]. Therefore, the gut microbiota offers a highly promising modifiable factor for tailoring dietary schemes for the purpose of any personalized dietary treatment [52], although alterations in the gut microbiota by means of diet are not straightforward, since responses will depend on the type of foods provided and it will vary between individuals. Moreover, the gut microbiota may not only be affected by diet, but it may also predict the response/non-response of specific dietary treatment on metabolic responses [62].

The complexity of the gut microbiota has been classified by different indicators such as alpha diversity (within-individual variation in gut microbiota composition) [63], via comparing the abundance ratio of Firmicutes to Bacteroidetes phyla, or via assessment of the bacterial genera as basis for enterotyping, i.e., stratification based on the gut microbiota. It has been suggested that three different enterotypes exist [64]. The first enterotype is common among people with a habitual Western diet rich in animal protein, and it typically has a higher abundance of Bacteroides spp. [64]. The second enterotype has a high abundance of Prevotella and is typically associated with people consuming diets rich in plant-based food [65]. Further, the use of the Prevotella/Bacteroides ratio as simplified proxy to characterize the enterotype has been suggested [33]. The third enterotype is characterized by a high abundance of the *Ruminococcus* genus within the phylum Firmicutes. Enrichment of *Ruminococcus* has been associated with irritable bowel syndrome (IBS) [66]. Grouping of gut microbiota composition into enterotypes has been suggested as an efficient means to estimate the individual responsiveness to dietary schemes such as a Nordic diet with regards to weight loss, postprandial glucose response and lipid lowering [30,33,34,67,68,69]. It should be recognized that the concept of enterotyping and its relevance to predict metabolic responses to specific diet is still in maturation. It is questionable whether the current enterotyping reflects the ratios and abundancies of individual bacterial species, and whether their existence is in homeostasis that would be characteristics for taken enterotype [70]. The current enterotyping does not take into account bacteria on the species level, nor their metabolic capacity that may be driven both individually or collectively and influence the host physiology in various ways. For example, within phyla Firmicutes and genus *Clostridium*, there are both pathogenic and synbiotic bacteria—*Clostridium difficile* causes diarrhea, while *Clostridium sporogenes* has an important metabolic role, synthesizing indole propionic acid from dietary tryptophan [71]. The complexity of the microbiota with over 1500 reference genomes catalogued, with constantly increasing numbers, sets a tremendous challenge for any classification or enterotyping approaches [53]. It is evident that there is currently no efficient method in use that could capture the complexity and variance of gut microbiota composition as a whole [70]. However, enterotyping has shown great potential as a predictor of response in the health effects of a Healthy Nordic Diet.

In a dietary intervention with either the New Nordic diet or the Average Danish Diet among 62 subjects with symptoms of metabolic syndrome (aged 18–65), it was shown that the participants formed two discrete groups by assessing their *Prevotella* spp. versus *Bacteroides* spp. ratio in fecal samples, rather than according to the actual abundancies of these bacterial taxa [33]. This ratio remained stable for 6 months, and dietary intervention with either the New Nordic diet or the Average Danish Diet did not seem to affect the ratio when the abundancies of 35 selected bacterial taxa were quantified prior and after the intervention. An interesting finding was that the enterotype appeared to have a larger impact on the total plasma cholesterol than diet. The Prevotella/Bacteorides ratio (P/B ratio) was associated with cholesterol levels. This finding suggested that microbiota affects cholesterol levels, and thus offers a potential means for the stratified control of cholesterol levels. In contrast to the study by Roager et al. [33], Eriksen et al. [34] found that a transient cholesterol reduction after the consumption of whole grain rye foods for 8 weeks was predicted by the base-line Prevotella enterotype, but not the other two enterotypes. The results corroborated with significantly higher propionate levels in the same enterotype. Furthermore, it has also been shown that P/B ratio predicts the ability of an individual to lose weight in response to adherence to a Healthy Nordic Diet, because study participants with a high P/B ratio were losing more body fat than those who had low P/B ratio during the intervention with diet high in fiber and whole grains [72].

The potential role of gut Prevotella abundance in the prediction of weight loss was further confirmed in another dietary intervention study, where overweight adults were on a 6 week intervention either with whole grain or refined grain products [31]. The weight loss was most evident in the group that was enterotyped as the high Prevotella group, thus offering possibility to utilize the enterotype in personalized stratification and also as a guide for a therapeutic avenue via modulation of the gut microbiota for enhanced responsiveness to weight management. Yet another short-term study (3 days) conducted with barley kernel bread showed that individuals who had higher Prevotella abundance responded to the dietary treatment and had improved glucose metabolism [30]. In a follow-up study it was defined that not the abundance of Prevotella per se but the Prevotella/Bacteroides ratio was the parameter capable of stratifying metabolic responders [73]. Furthermore, Prevotella has also been associated inversely with clinical parameters of obesity related low-grade inflammation, including lipopolysaccharide and high-sensitive C-reactive Protein (hs-CRP), as was reviewed recently [74]. Therefore, the abundance of Prevotella spp. via enterotyping or any other proxy reflecting the relative proportion of Provotella in gut microbiota composition seems a promising approach not only predicting the dietary responsiveness to a fiber-rich Healthy Nordic Diet but also as a therapeutic approach for weight loss via modulation of gut microbiota composition [75].

Collecting fecal samples for assessment of gut microbiota composition is demanding for study subjects and the methodologies to address gut microbiota composition are rather expensive. Moreover, gut microbiota activity rather than composition may be more informative in relation to health effects [76]. Shot-gun metagenome sequencing offers a genomic-based approach to address gut microbiota functionality, but it is expensive and requires extensive bioinformatics processing. Metabolomics on fecal samples have been shown to reflect not only gut microbiota composition to a great extent but also its metabolic activity [76,77] and many fecal metabolites can also be detected in blood samples, therefore offering a straightforward analytical alternative [77].

## 6. Effects of a Healthy Nordic Diet on the Gut Microbiota and Derived Molecules

The impact of the gut microbiota on the human host is mostly mediated via molecules originating from the diet or endogenous metabolism and which are processed in the metabolism of the gut bacteria. These compounds enter the circulation and subsequently the target organs and they offer a possibility to measure the biochemical activity of the gut microbiota and, to some extent, its composition [77,78]. As such, they may serve as indicators that reflect gut microbiota composition, specific food intakes or the interaction between diet and gut microbiota related to specific health-related metabolic traits. The Healthy Nordic Diet contains several dietary sources that have long been known to interact with the gut microbiota and give rise to specific molecules which have been associated with health outcomes. Such molecules derived from gut microbiota interaction with whole grains, berries, and fish, may result in changes in circulating metabolites that have potential to be utilized as a monitoring panel for responsiveness to diet with the input of the gut microbiota.

One of the most crucial roles of the gut microbiota is to harvest the energy from complex dietary carbohydrates that are not amenable for human endogenous metabolism directly, but only via the utilization of short-chain fatty acids (SCFAs) that are produced from their breakdown by the gut microbiota in colon [79]. Dietary fiber is often discussed as an important dietary factor to shift the gut microbiota towards a healthier direction. Only small alterations in the gut microbiota overall have been observed in response to a high whole grain diet [80,81], but it has been shown that interventions with whole grains result in an increase in SCFA-producing *Lachnospira* spp. when compared with a control group consuming refined grains [82]. However, the increase in SCFA-producing bacteria has not been reflected by more than marginal changes in SCFA concentrations in stool samples and in systemic blood samples [81] after whole grain interventions. It has been reported that the Prevotella enterotype has a more efficient production of SCFA [83]. SCFA concentrations in circulation reflect, to some extent, both the quantity and quality of dietary fiber and the gut microbiota carbohydrate-degrading capacity. SCFA could be potential mediators of at least part of the beneficial effects observed for high intake of fiber diet, such as the Healthy Nordic Diet, on cardiometabolic outcomes [84].

One of the compounds derived from the gut microbiota in interaction with a specific diet, which is currently under intensive focus, is Trimethylamine N-oxide (TMAO). It is produced by the gut microbiota from dietary choline and carnitine sources and is linked with adverse effects on cardiometabolic health [85]. However, TMAO can also be directly absorbed from dietary sources, mainly fish [86]. Fish has been long linked with positive health effects including protection from cardiovascular diseases [87]. For example a recent report on the European Prospective Investigation Into Cancer and Nutrition (EPIC)-Oxford study concluded that fish eaters and vegetarians had 13% lower rates of ischemic heart disease than meat eaters [88] and dietary patterns rich in fish, such as the Nordic diet and Mediterranean diet, are among most stably reported health-beneficial dietary schemes [3]. This is in controversy with the fact that after the consumption of fish, the plasma circulating levels of TMAO already rise 15 min after a meal, reaching approximately 50 times higher circulating concentrations of TMAO than beef or egg consumption [86]. This apparent controversy has raised questions which may not be resolved until a more comprehensive understanding of the metabolism of diet-derived TMAO is disclosed [89]. In addition to promoting atherosclerosis, TMAO has also been linked with cholesterol/bile acid metabolism. Interestingly, Roager et al. [81] found that the Prevotella enerotype has higher levels of cholesterol despite the dietary intervention group and it has surprisingly been found that Prevotella has been associated with higher levels of TMAO, despite the fact that the Prevotella enterotype is typically associated with a plant-based fiber-rich diet [90]. However, data on this aspect is still scarce, and investigations on larger populations are required to elucidate whether the Prevotella enterotype is characteristic of high cholesterol and TMAO, and where the metabolic linkage between this and cardiovascular health is. Notable, in a recent investigation on various trimethylated compounds during dietary intervention with the Nordic diet, TMAO was increased in circulation likely due to the fish intake, similarly as biomarkers related to whole grain intake (e.g., pipecolic acid betaine), but it had an opposite association with various clinical measurements including cholesterol [46]. Whilst the precise metabolic role of TMAO yet requires revisiting, it is in any case a potential candidate compound to be included in the metabolite repertoire for monitoring the individual response to diet, since it is linking dietary habits, gut microbiota, and metabolic health.

One of the most commonly detected compounds of microbiota origin, and often reported in studies where Healthy Nordic Diets have been investigated, is hippuric acid. Hippuric acid originates from the gut microbiota-mediated breakdown of dietary phenolic compounds into benzoate and glycine conjugation in the liver with hippuric acid as the end product [91]. In spite of its ease of measurement by both mass spectrometry and NMR-based analytical approaches, it has so far gained relatively little attention as a potential biomarker for diet/microbiota/health responsiveness, although there are a few studies clearly demonstrating its possibilities for this purpose. De Mello et al. [92] reported that hippuric acid was increased in fasting plasma after dietary intervention with bilberries, and was associated with improved glucose homeostasis [92]. Another study reported that hippuric acid was associated with a lower risk of metabolic syndrome in the Twins UK cohort, and interestingly, was also related to higher microbiome diversity (Shannon diversity) [93]. More precisely, hippuric acid was positively associated with the species *Faecalibacterium prausnitzii,* a butyrate-producing microbial species that has often been linked with lower inflammatory markers [74] and suggested as potential therapeutic agent in gut transplantation [94].

In addition to the compounds discussed above, several other microbiota-produced metabolites have been consistently linked with Nordic dietary habits and gut microbiota function, such as fish-derived furan fatty acids, trimethylated compounds, and indole-metabolites. These compound groups have not yet been studied in detail, but they do offer interesting candidates for further studies.

## 7. Conclusions and Future Directions

A Healthy Nordic Diet contains a combination of food items such as whole grains, root vegetables, fish and berries contributing to beneficial health effects as evidenced from observational studies and to some degree from clinical studies. The analysis of the metabolic impact of whole diet possesses great challenge, as it encompasses a focus on the multitude of food items, hundreds of diet-derived compounds, and numerous endogenous metabolic processes, driven by individual characteristics. The current use of dietary and effect biomarkers derived from metabolomics as well as gut microbiota-based entrotyping and metabolites reflecting microbiota x diet interactions and their current use in studies on a Healthy Nordic Diet and health have been summarized schematically below in Figure 1. We also highlighted areas where there are currently gaps in the use of such biomarkers (Figure 1).

In conclusion, metabolomics has proven a useful technique to derive food intake biomarkers of the Healthy Nordic Diet as well as address metabolic alterations induced by such a diet. Moreover, the gut microbiota interacts both with dietary compounds as well as human metabolism, adding yet another level of complexity to the whole picture. Metabolites derived from the gut microbiota in interaction with a Healthy Nordic Diet have been identified, but more studies are needed. In order to improve the efficacy of the Healthy Nordic Diet, individuals may need to follow personalized nutrition strategies as indicated in recent studies showing the larger effects of the Healthy Nordic Diet intervention among subjects with certain gut microbiota and glucose/insulin profiles before the intervention [69].

Results from post hoc, explorative studies using data and samples from randomized controlled trials (RCTs) on the Healthy Nordic Diet [32,68,95] or specific food components [30,96] thereof, have proven useful to elucidate the factors of importance to explaining/predicting the response of a Healthy Nordic Diet or important food components thereof. However, so far, no controlled intervention study has evaluated the effects of a personalized approach taking such factors into account vs. a general Healthy Nordic Diet and it therefore remains to be elucidated to what extent personalized nutrition strategies vs. general advice based on a Healthy Nordic Diet has advantages and whether it is feasible at a large scale [97,98,99].

Most recently, approaches taking into account individual responsiveness to the Healthy Nordic Diet dependent on gut microbiota composition seem a highly promising avenue for future personalized nutrition guidance to enhance the dietary mediated prevention of non-communicable diseases. The monitoring of the gut-derived metabolites in circulation using metabolomics techniques may offer a yet largely unexplored possibility to better reflect gut microbiota functionality, but the suitable and reliable pattern of such metabolites related to improved response to a Nordic diet and its implications for health is yet to be defined in coming studies. Furthermore, future studies are under way to explore whether metabolomics could be used to discover biomarkers of metabotypes, i.e., groups of individuals with a similar metabolic phenotype [100], associated with a differential response to a Healthy Nordic Diet. Currently, no study has been reported where a Healthy Nordic Diet or components thereof have been included in algorithms that tailor a diet for any outcome at the individual level. This remains a big challenge for research to come.

## Figures and Tables

**Figure 1 nutrients-12-00027-f001:**
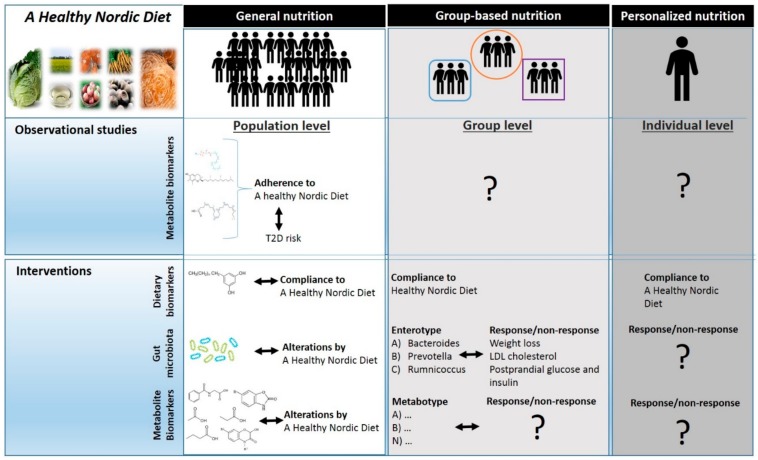
The reported and yet unknown applications of metabolite biomarkers reflecting effects, dietary intake/adherence of a Healthy Nordic diet in observational and intervention studies at the population-, group- and individual levels. Moreover, the effects of a Healthy Nordic Diet on the gut microbiota as well as the use of the gut microbiota to assess differential responses for intertypes (group level) or at individual level. LDL-low density lipoprotein; T2D-type 2 diabetes.

**Table 1 nutrients-12-00027-t001:** Putative biomarkers reflecting the intake of a Healthy Nordic Diet derived from dietary intervention studies with Healthy Nordic Diets and observational studies where adherence to Healthy Nordic Diet-related food indices were measured.

Type of Study	Participants	Duration/Follow-Up	Foods/Dietary Instrument	Sample/Analysis	Metabolites That Differed between Diets/Adherence to Indices	Type of Biomarkers	Ref.
Intervention study, parallel	In total, 181, overweighed + one MetS risk factor	6 months	New Nordic diet ^1^ vs. habitual Danish diet/repeated weighed food records and foods provided in an intervention shop	24-Urine/UPLC-QTOF-MS ^1^	In total, 52 metabolites explained differences between the diets, but trimethylamine N-oxide, hippuric acid, hydroquinone-glucuronide, (2-oxo-2,3-dihydro-1H-indol-3-yl)acetic acid and 3,4,5,6-Tetrahydrohippuratewere indicative of New Nordic Diet.	Compliance biomarkers	[47]
Intervention study, parallel	In total, 161, overweighted + one MetS risk factor	6 months	New Nordic diet^1^ vs. habitual Danish diet/repeated weighed food records and foods provided in an intervention shop	Fasting plasma/GC–MS ^1^	In total, 33 metabolites differentiated between groups but 3-hydroxybutanoic acid, erythritol, 2-hydroxybenzoic acid, aspartic acid, 2,3,4-trihydroxybutanoic acid, xylitol, N-acetylaspartic acid, 2,5-dimethoxyphenylpro-pionic acid, and palmitoleic acid were indicative of the New Nordic Diet	Effect biomarkers, biomarkers of weight loss, seasonality biomarkers and dietary biomarkers	[41]
Intervention study, parallel	In total, 161, overweighted + one MetS risk factor	6 months	New Nordic diet ^2^ vs. habitual Danish diet/repeated weighed food records and foods provided in an intervention shop	Fasting plasma/UPLC-QTOF-MS	Food intake-related metabolites included theobromine (chocolate), proline betaine (citrus), products of food heating (a cyclic dipeptide, i.e., cyclo(pro-val)), fish (TMAO),products of animal protein metabolism (a tryptophan metabolite, indolelactic acid), and novel markers of other food groups (i.e., pipecolic acid betaine and prolyl-hydroxyproline). Endogenous metabolites shifted included butyryl carnitine, 2-hydroxy-3-methylbutyrate, specific phospholipids and plasmalogens	Effect biomarkers, biomarkers of weight loss, seasonality biomarkers and dietary biomarkers	[42]
Intervention, parallel	In total, 106 men and women with MetS	12 weeks	(1) whole grains + fatty fish + billberries; (2) whole grains; (3) refined wheat/weighed food records	Fasting plasma/UPLC-QTOF-MS	Glucuronidated alk(en)- ylresorcinols were correlated with whole grains, diet (2) were higher in 3-carboxy-4-methyl-5-propyl-2-furanpropanoic acid (CMPF), hippuric acid, and various lipid species incorporating polyunsaturated fatty acids	Food intake biomarkers and effect biomarkers	[43]
Intervention, parallel, multicenter	In total, 213 (166 completed) men and women with metabolic syndrome	18–24 weeks depending on study center	Healthy Nordic Diet ^3^ including whole grain products, berries, fruit and vegetables, rapeseed oil, three fish meals per week, low-fat dairy products and avoidance of sugar-sweetened products vs. control diet comprising and average Nordic diet/food records	Fasting plasma/LC-QQQ-MS ^1^	Pipecolic acid betaine (PAB) was significantly higher in the Healthy Nordic Diet group than in the control group at the end of the intervention. No other metabolites differed significantly	Effect biomarkers	[41]
Observational, nested case-control study	In total, 421 case-control pairs of healthy men and women		Adherence to the Healthy Nordic Food Index and Baltic Sea Diet Score; FFQ	Fasting plasma/ UPLC-QTOF-MS	In total, 31 metabolites were associated with BSDS and/or HNFI. Five metabolites were associated with both indices: docosahexaenoic acid (DHA), lysophatidylethanolamine (lysoPE 22:6), γ-tocopherol, and two unknown metabolites	Food intake biomarkers and effect biomarkers	[44]

^1^ Abbreviations: ultra-high performance liquid chromatography-quadrupole time-of-flight mass spectrometry (UPLC-QTOF-MS); Gas Chromatrography-Mass Spectrometry (GC-MS); Liquid Chromatography-triple Quadrupole Mass Spectrometry (LC-QQQ-MS); Baltic Sea Diet Score (BSDS); Healthy Nordic Food Index (HNFI). ^2^ Food items from 15 food groups such as vegetables, fruits, berries, whole grains, nuts and seafood products were included (see [50]). ^3^ For a detailed description of the diets, see Uusituppa et al. [10].

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
