# Peer review of "Biomarkers of a Healthy Nordic Diet—From Dietary Exposure Biomarkers to Microbiota Signatures in the Metabolome"

_nutrients, 2019, doi:10.3390/nu12010027_

Round 1

Reviewer 1 Report

The authors aim to summarise and provide further interpretation on an interesting but complex topic of biomarker signatures based on metabolomics and microbiota analyses for the healthy Nordic diet. Just as the authors conclude, many factors need to be considered, which limits the possibilities to draw firm conclusions.

I think the authors have done a good job in summarising the available literature on this topic. However, while reading the paper it is still difficult to see the full picture. To my opinion the manuscript may benefit from some visual support. The table is the only ‘non-textual’ information, and that table itself is too large and difficult to comprehend.

I have the following suggestions:

1) The authors should add a (large) figure as a summary, to illustrates the potential pathways based on the available metabolomics and the microbiota data, to indicate what is known and where the gaps are.

2) The table should be revised, maybe delete some of the columns or split in two or more tables. As it is now, the table is not readable.

Author Response

Response to reviewer 1:

The authors aim to summarise and provide further interpretation on an interesting but complex topic of biomarker signatures based on metabolomics and microbiota analyses for the healthy Nordic diet. Just as the authors conclude, many factors need to be considered, which limits the possibilities to draw firm conclusions.

I think the authors have done a good job in summarising the available literature on this topic. However, while reading the paper it is still difficult to see the full picture. To my opinion the manuscript may benefit from some visual support. The table is the only ‘non-textual’ information, and that table itself is too large and difficult to comprehend.

Response: We thank the reviewer for this suggestion and we have therfore now prepared a figure to illustrate the scope adressed in the paper. See Figure 1.

I have the following suggestions:

1) The authors should add a (large) figure as a summary, to illustrates the potential pathways based on the available metabolomics and the microbiota data, to indicate what is known and where the gaps are.

Response: Thank you! This is a great suggestion and we have now added Figure 1.

2) The table should be revised, maybe delete some of the columns or split in two or more tables. As it is now, the table is not readable.

Response: We agree with the reviewer and therfore we have now revised the table. We have put it as landscape and deminished the fonts.

Reviewer 2 Report

In this review paper by Landberg and Hanhineva, the authors report the state of the art about the study of the health effects of Nordic Diet from a mechanistic point of view, mainly conducted using metabolomics and multi-omics approaches. As these type of studies are relatively recent, the data available in literature are not massive up to now, hence this work could be regarded as a perspective on the future direction of the research in this field.

I think that the authors performed a good work. The most recent literature is properly reviewed, and the data are reported in a clear and ordinate way and are correctly contextualized. Some minor issues should be re-checked and corrected by the authors.

First of all, I suggest to carefully read the manuscript and detect and correct minor spelling and grammatical errors throughout the text Line 203: What “NND” and “ADD” stand for? Please explain in the text Line 218: It is not clear what the “4 different LC modes” stand for. Please, explain in the text. Line 224: what does “CMPF” mean? As stated before, please report the meaning of each acronym at its first appearance in the text. Table 1: although the table is well done, I suggest to report only the metabolites that were identified and related to the specific intervention in the column “Metabolites”.

Author Response

In this review paper by Landberg and Hanhineva, the authors report the state of the art about the study of the health effects of Nordic Diet from a mechanistic point of view, mainly conducted using metabolomics and multi-omics approaches. As these type of studies are relatively recent, the data available in literature are not massive up to now, hence this work could be regarded as a perspective on the future direction of the research in this field.

I think that the authors performed a good work. The most recent literature is properly reviewed, and the data are reported in a clear and ordinate way and are correctly contextualized. Some minor issues should be re-checked and corrected by the authors.

First of all, I suggest to carefully read the manuscript and detect and correct minor spelling and grammatical errors throughout the text.

Response: We thank the reviewer for positive feedback. We have now carefully red the paper and corrected language mistakes.

Line 203: What “NND” and “ADD” stand for?

Response: We have now explained NND = New Nordic Diet and "ADD" =Average Danish Diet. Thank you.

Please explain in the text Line 218: It is not clear what the “4 different LC modes” stand for. Please, explain in the text.

Response: Thank you, this has now been explained (reversed phase chromatography and HILIC (normal phase chromatography) in positive and negative inoization modes).

Line 224: what does “CMPF” mean? As stated before, please report the meaning of each acronym at its first appearance in the text.

Response: Thank you. We have now explained this and other abbreviations first time used. CMPF= 3‐carboxy‐4‐methyl‐5‐propyl‐2‐furanpropanoic acid 

Table 1: although the table is well done, I suggest to report only the metabolites that were identified and related to the specific intervention in the column “Metabolites”.

Response: Thank you for a good suggestion, we agree. Now changed.